# Subspecialty Second-Opinion in Multiple Myeloma CT: Emphasis on Clinically Significant Lytic Lesions

**DOI:** 10.3390/medicina56040195

**Published:** 2020-04-23

**Authors:** Alberto Stefano Tagliafico, Liliana Belgioia, Alessandro Bonsignore, Federica Rossi, Giulia Succio, Bianca Bignotti, Alida Dominietto

**Affiliations:** 1IRCCS Ospedale Policlinico San Martino, 16132 Genova, Italy; liliana.belgioia@unige.it (L.B.); alessandro.bonsignore@unige.it (A.B.); giulia.succio@hsanmartino.it (G.S.); bignottibianca@gmail.com (B.B.); alida.dominietto@hsanmartino.it (A.D.); 2Department of Health Sciences (DISSAL), University of Genoa, 16132 Genoa, Italy; federossi0590@gmail.com; 3Department of Experimental Medicine (DIMES), University of Genoa, 16132 Genoa, Italy

**Keywords:** multiple myeloma, computed tomography, second-look, lytic lesions, bone, staging

## Abstract

*Background and objectives*: In order to increase the accuracy of lytic lesion detection in multiple myeloma, a dedicated second-opinion interpretation of medical images performed by subspecialty musculoskeletal radiologists could increase accuracy. Therefore, the purpose of this study is to evaluate the added value (increased accuracy) of subspecialty second-opinion (SSO) consultations for Computed Tomography (CT) examinations in Multiple Myeloma (MM) patients undergoing stem cell transplantation on standard computed tomography with a focus on focal lesion detection. *Materials and Methods*: Approval from the institutional review board was obtained. This retrospective study included 70 MM consecutive patients (mean age, 62 years ± 11.3 (standard deviation); range, 35–88 years) admitted in the last six years. Pre-transplant total-body CT (reported by general radiologists) was the only inclusion criteria. Each of these CT examinations had a second-opinion interpretation by two experienced subspecialty musculoskeletal (MSK) radiologists (13 years of experience and 6 years of experience, mean: 9.5 years), experts in musculoskeletal radiology and bone image interpretation with a focus on lytic lesions. *Results*: Per lesion intra- and inter-observer agreement between the two radiologists was calculated with K statistics and the results were good (K = 0.67: Confidence Inteval (CI) 95%: 0.61–0.78). When the initial CT reports were compared with the re-interpretation reports, 46 (65%) of the 70 cases (95% CI: 37–75%) had no discrepancy. There was a discrepancy in detecting a clinically unimportant abnormality in 10/70 (14%) patients (95% CI: 7–25%) unlikely to alter patient care or irrelevant to further clinical management. A discrepancy in interpreting a clinically important abnormality was registered in 14/70 (21%) patients for focal lesions. The mean diameter of focal lesions was: 23 mm (95% CI: 5–57 mm). The mean number of focal lesions per patient was 3.4 (95% CI). *Conclusions*: subspecialty second-opinion consultations in multiple myeloma CT is more accurate to identify lesions, especially lytic lesions, amenable to influence patients’ care.

## 1. Introduction

Multiple myeloma (MM) is a hematologic disorder characterized by an excessive production of the immunoglobulin M component of plasma cells. In MM, the bone lesions of myeloma are determined by the proliferation of cells from a single clone. Then, osteoclasts are activated and destroy the bone [1]. MM, known with the abbreviation CRAB (hyperCalcemia, Renal failure, Anaemia, and lytic Bone lesions) is a cytogenetically heterogenous disorder of clonal plasma cells [1]. The extent of the bone disease negatively influences patients’ quality of life, increasing both morbidity and mortality. The detection of lytic bone lesions on imaging separates asymptomatic from symptomatic MM patients, even if no clinical symptoms are present [1,2,3,4]. Medical imaging is pivotal in the management of patients with MM. Imaging is used to detect bone lesions, to predict the risk of early progression from smoldering MM (sMM) to active MM, to identify extra-medullary disease and to identify the sites of possible pathologic fractures or neurologic complications [3]. In patients with a recent diagnosis of MM, focal lesions detected with Magnetic Resonance Imaging (MRI) or Computed Tomography (CT) or Positron Emission Tomography (PET)/CT are important for correct treatment and for prognosis [3]. In MM, “focal lesions” detected by MRI should not be confused with “lytic lesions” detected by CT. Indeed, the detection of at least one lytic lesion is a negative prognostic factor for patients with MM [2,5,6]. In 2014, the International Myeloma Working Group (IMWG) updated the definition of MM: the presence of at least one lytic lesion detected not only by conventional radiography but also by CT, WBLDCT, or PET/CT was included in the definition [7]. The incorporation of imaging modalities such as CT and PET/CT is recommended (grade A) according to the recent literature [3]. However, in MM patients, differentiation between a focal and a diffuse pattern on CT is still difficult even with Radiomics [6]. To increase the accuracy in lytic lesion detection, a dedicated second-opinion interpretation of medical images performed by subspecialty musculoskeletal radiologists could be more accurate. 

In many centres, consultation and second-opinion interpretation of medical images by subspecialty radiologists are routinely performed [8,9,10,11,12]. Therefore, the purpose of this study is to evaluate the increased detection of focal lesions and other radiological findings of subspecialty second-opinion (SSO) consultations for CT examinations in MM patients undergoing stem cell transplantation on standard computed tomography. 

## 2. Materials and Methods

Approval from the institutional review board was obtained (003REG2019). All patients signed a written, informed consent form for retrospective research purposes, before CT examination. SSO was applied to CT data collected in the clinical workup and did not influence patient care in any way because the study was made retrospectively. 

### 2.1. Inclusion Criteria

This retrospective study evaluated *n* = 70 consecutive patients (mean age, 62 years ± 11.3 (standard deviation); range, 35–73 years) treated at the IRCCS Policlinico San Martino Hospital (Genoa, Italy) for MM in the last six years. Pre-transplant total-body CT with minimal technical standard (Table 1) available in the Hospital Picture Archiving and Communication System (PACS) or available in DICOM format from CT acquired outside the hospital were the only inclusion criteria—the initial CT reading was done by general radiologists with no known formal (ESSR Diploma, track record in MSK radiological activities) or informal (staff rounds, reports of specialized MSK exams) specialized experience in MSK radiology. 

### 2.2. Study Design

CT examinations were studied with a second-opinion interpretation by two experienced MSK radiologists (A.T. 13 years of experience, F.R. 6 years of experience, mean: 9.5 years). The two radiologists evaluated the CT examination blindly and in different sessions. To avoid reading biases, an independent medical student was enrolled as data controller (DC). The DC checked that second-opinion interpretation was done after removing all the information of the original CT. The original report was removed. In addition, the DC made sure not to include a CT examination when the radiologists had already been involved in image re-interpretation. The use of a DC has already been explored in the literature [12]. 

Second-opinion consultation was made independently by using a 3-point scoring system. The scoring system is similar to a scoring system already published and now adapted to MM patients [12]: 1, no discrepancy; 2, discrepancy in detecting an unimportant abnormality (e.g., interpreting a bone infarct as a bone island, osteophytes, disc degeneration, old vertebral collapse, not neoplastic or clearly benign bone lesions); 3, discrepancy in interpreting an important abnormality (e.g., interpreting the presence of a lytic lesion >5 mm). Lytic bone lesions, size or number, non-lytic lesions, extramedullary manifestations and osteonecrosis (only if not detected by general radiologists), and fractures were considered. The clinically important differences were defined as those likely to change patient care or diagnoses according to suggestions given by our clinician on a per patient analysis (for example, a lytic lesion in a CT reported as negative at initial reading). For example, a lytic lesion could be used to stage the disease according to the Durie and Salmon PLUS staging system. After per lesion intra- and inter-observer agreement calculation, reports were re-evaluated together when their scorings were discordant. Discrepancies, mainly lytic lesions, that were significant enough to warrant a change in diagnosis, prognosis, invalidity (for medico-legal implications) or treatment or referral (e.g., orthopedic surgeon, radiation oncologist specialist) were recorded.

### 2.3. Reference Standard

For this study, radiologists’ consensus was the best feasible reference standard available [2,6] because biopsy is not always available for all suspicious areas on CT. The best valuable comparator, or reference standard (BVC), was constructed as described elsewhere [5,6,7,13,14]. One hematologist, two radiologists and one radiation oncologist, all with > 10-year clinical experience, reviewed CT, MRI and PET/CT examinations and clinical follow-up for clinical significance. True positive or negative examinations were defined by lesion progression or by new lesions on follow-up imaging, or lesion response with therapy, and evolution of biologic parameters. False positive examinations were defined by an absence of new lesions on follow-up imaging studies. False negative examinations were defined by the failure of lesion detection [15]. Diffuse bone marrow infiltration in the skeleton was recorded according to Staebler et al. (lesions <5 mm, not osteoporosis) [15]. Focal pattern was defined as the presence of at least one >5 mm focal or lytic lesion. The presence of at least one focal or lytic lesion was considered relevant because it is a prognostic factor MM [5,6]. 

### 2.4. Statistical Analysis

(1) Per lesion intra- and Inter-observer agreement between the two radiologists was calculated with K statistics. *p* values below 0.05 were considered statistically significant. Agreement was assessed according to Altman [16] and adapted from Landis and Koch [17]. Values of 0.81–1.00 indicated very good agreement, 0.61–0.80 indicated good agreement, 0.41–0.60 indicated moderate agreement, 0.21–0.40 indicated fair agreement, and 0.20 or lower indicated poor agreement; 

(2) Statistical comparisons of rates were performed using a chi-square test with Bonferroni corrections. Statistical tests were done using statistical software (STATA MP, StataCorp, 4905 Lakeway Dr, College Station, TX, USA and MedCalc).

## 3. Results

Intra- and Inter-observer agreement between the two radiologists was calculated with K statistics and the results were good (K = 0.67: IC 95%: 0.61–0.78) in scoring the discrepancies between subspecialized second-opinion consultations and standard CT reports, but consensus scores were used for further analysis as planned in the study protocol. Overall scores of subspecialized second-opinion consultations versus outside reports are summarized in Table 2.

As reported in Table 2, when the initial CT reports were compared with the re-interpretation reports, 46 (65%) of the 70 cases (95% CI: 37–75%) were graded 1, no discrepancy. There was a discrepancy in detecting a clinically unimportant abnormality in 10/70 (14%) patients (95% CI: 7–25%) unlikely to alter patient care or irrelevant to further clinical management. A discrepancy in interpreting a clinically important abnormality (e.g., interpreting the presence of a lytic lesion >5 mm) was registered in 14/70 (21%) patients. As shown in Table 3, the majority of discrepancies that were clinically significant (Score Category 3) were due to significant focal lesion detection discrepancies. The mean diameter of all detected focal lesions was: 23 mm (95% CI: 5–57 mm). The mean number of focal lesions per patient was 3.4 (range: 0-20; 95% CI:1.1–4.7). As a whole, *n* = 60 patients had focal lesions and *n* = 10 had none. In *n* = 14 patients without detected lesions by the initial report SSO found “new” lesions, thus potentially changing further treatment planning. 

## 4. Discussion

In radiological clinical practice, it is quite common to have dedicated subspecialty second-opinion consultations, especially in tertiary academic centres with tumor board meetings, often known as disease management teams. However, we were not able to find the relevant literature regarding subspecialty second-opinion consultations in multiple myeloma CT. Indeed, there is growing interest in the evaluation of bone status in MM due to the increasing evidence that the presence of certain bone marrow patterns may be useful to stage and predict the outcome of MM [5,6,7,13,18,19]. In addition, there is a growing interest in the evaluation of lytic lesions due to their possible influence on prognosis [5]. For example, Rasche et al. [5] investigated the prognostic value of focal lesion size in 404 transplant-eligible, newly diagnosed, MM patients with Magnetic Resonance Imaging. The authors [5] used a diffusion-weighted sequence to identify the presence of multiple large focal lesions. They found that focal lesions are strong prognostic factors. According to Rasche et al. [5], of patients with at least three large focal lesions with a product of the perpendicular diameters >5 cm, two were associated with poor progression-free survival and overall survival. This pattern was seen in 13.8% of patients and was independent of the Revised International Staging System [5]. In 2010, Hillengass et al. [18], using Whole Body Magnetic Resonance Imaging, found that the presence of focal lesions is the strongest adverse prognostic factor for progression. CT and PET/CT are now highly recommended in MM evaluation [3] and the lytic lesion assessment in MM is difficult [6]. Therefore, the focus of the present study is to improve the detection and characterization of clinically significant lytic lesions. In the past, discrepancies between reports by radiologists at different levels of training and radiologists at different clinical settings had discrepancy rates from 0.1% to 15% [12,20]. Compared to the published literature, we found that the discrepancy rate in interpreting a clinically important abnormality (e.g., interpreting the presence of a lytic lesion >5 mm) was 21% (14/70 patients), which is slightly higher than the literature data. However, we do not have any MM-related data for comparison, but only data derived from other pathological conditions. Our results highlight the necessity and the potential benefit of a subspecialty second-opinion consultation in multiple myeloma CT, in order to avoid medico-legal consequences. Furthermore, the main pathological finding that determined discrepancies was the presence of a lytic lesion. The lytic lesion of MM could be difficult to detect, especially when the diameter was between 5 and 10 mm and when located in an osteoporotic and degenerated vertebral body. In these cases, the experience of dedicated MSK radiologists could be important. Some small lytic lesions, for example, could be confused with Schmorl nodes, also referred to as intravertebral disc herniations. This study has several limitations. We acknowledge that some CTs were not primarily acquired to evaluate and detect focal lesions, therefore it is likely that these focal abnormalities were under-reported. Perhaps a more focused clinical indication before CT acquisition and report could improve focal lesion detection. Proper education of radiologists reporting MM radiological evaluation, could improve the quality of the report further. There is no clear instruction at present in the primary report for how to categorize disease entities regarding clinical relevance, therefore some of those related to MM may be overlooked or even overestimated. In addition, second-look interpretations and primary readings have been performed in different environments with different clinical priorities and different levels of expertise. Furthermore, in certain radiological environments, there is an emphasis on the quantity of work produced, which is easier to measure than the quality of interpretation [11]. Another limitation is that the scenario where expert MSK radiologists are present to reevaluate the CTs of MM patients is difficult to propose. Indeed, subspecialty radiologists practice in large and academic departments and are rare in smaller centres. In many developing countries, only general radiologists are available and imaging interpretation is sometimes performed by physicians with very limited training [21]. Finally, no correlation between discrepancies and the clinical outcome of MM patients was possible to report due to the limited number of patients, the retrospective nature of the study, and the fact that the presence of focal lesion is not the only determinant of poor prognosis.

## 5. Conclusions

In conclusion, our study demonstrated that subspecialty second-opinion consultation in multiple myeloma CT could identify lytic lesions, previously missed, amenable to influence patients’ care.

## Figures and Tables

**Table 1 medicina-56-00195-t001:** Minimal and standard Computed Tomography Technical parameters for inclusion.

Number of Detector Rows	16 or More up to 128
Minimum Scan coverage	Skull base to femur
Tube voltage(kV)/time-current product (mAs)	120/50–70, adjusted as clinically needed
Reconstruction convolution kernel	Sharp, high-frequency (bone) and smooth (soft tissue). Middle-frequency kernel for all images are adjusted by the radiologist as deemed necessary
Iterative reconstruction algorithms	Yes (to reduce image noise and streak artefacts)
Thickness	≤5 mm
Multiplanar Reconstructions (MPRs)	Yes (sagittal, coronal and parallel to long axis of proximal limbs)
Matrix, Rotation time, table speed, pith index	128 × 128, 0.5 s, 24 mm per gantry rotation, 0.8

**Table 2 medicina-56-00195-t002:** Consensus Scores of Subspeciality Second-Opinion Consultation Versus Standard CT Interpretation.

Discrepancy Score Category	No. (%) of Examinations
1, no discrepancy.	46 (65%)
2, discrepancy in detecting a clinically unimportant abnormality (e.g., a missed case of mild degenerative disease, interpreting a bone infarct as a bone island).	10 (14%)
3, discrepancy in interpreting a clinically important abnormality (e.g., interpreting the presence of a lytic lesion >5 mm or the presence of osteonecrosis or vice versa).	14 (21%)
Total	70 (100%)

**Table 3 medicina-56-00195-t003:** Disease Category Versus Discrepancy Rates.

Disease Category	Discrepancy Score Category 1	Discrepancy Score Category 2	Discrepancy Score Category 3
Focal Lesion Detection	46	-	14
Diffuse Pattern	17	4	-
Osteonecrosis	-	1	-
Number of Focal Lesion	-	6	-

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
