# Peer review of "Subspecialty Second-Opinion in Multiple Myeloma CT: Emphasis on Clinically Significant Lytic Lesions"

_medicina, 2020, doi:10.3390/medicina56040195_

Round 1

Reviewer 1 Report

Tagliafico et al. described how an accurate revision of radiological assessement by TC in patients with MM, revealed no discrepancy (65%) in the results by a panel of muscoloskeletal radiologist experts. These results are a good point for a better caracterizione and a better correlation with clinical outcome of those patients who where misdiagnosticated. However, They have to make:

Major revision:

  • Median age 62 range 35-88 in trasnplant patients: how is it possible to transplant > 80 Y patiens? Can you verify?
  • In the abstract is not clear what kind of discrepancies you have: focal lesions? osteonecrosis? diffuse signal?
  • The major point is that ther is no correlation between discrepancies and clinical outcome of MM patients: for example a patients with a focal lesion that was not evaluated at baseline did evolve with mass or other lesions? did progress or die early?
  • English should be revised in some parts

Minor revision:

Line20 and line70: please rephrase "the accuracy in assessing accurately", this sentence is redundant

Line 49: please rephrase: " MM is a cytogenetically heterogenous disorder of clonal plasma cells known with the abbreviation "CRAB"", this sentence is not cleare, better is: " MM, known with the abbreviation "CRAB,  is a cytogenetically heterogenous disorder of clonal plasma cells".

Author Response

Comments and Suggestions for Authors

Tagliafico et al. described how an accurate revision of radiological assessement by TC in patients with MM, revealed no discrepancy (65%) in the results by a panel of muscoloskeletal radiologist experts. These results are a good point for a better caracterizione and a better correlation with clinical outcome of those patients who where misdiagnosticated. However, They have to make:

Major revision:

  • Median age 62 range 35-88 in transplant patients: how is it possible to transplant > 80 Y patiens? Can you verify?

- Page 2 Line 86: indeed it is a mistake as shown by mean age and SD (maximum age 73). We corrected accordingly and thank you for noticing.

  • In the abstract is not clear what kind of discrepancies you have: focal lesions? osteonecrosis? diffuse signal?

- Page 1 Line 37: we highlighted that the discrepancy was due to focal lesion (in the abstract, details are in the tables).

  • The major point is that ther is no correlation between discrepancies and clinical outcome of MM patients: for example a patients with a focal lesion that was not evaluated at baseline did evolve with mass or other lesions? did progress or die early?

- We completely agree with this observation. It would be interesting to assess if a focal lesion that was not evaluated at baseline did evolve with mass or other lesions or the patient progress or die early. Unfortunately the study design, retrospective and "cross-sectional-like" does not allow a thorough assessment of this point. We have few patients with defined outcome (relapse or death) to differentiate if the group of patients with a focal lesion missed has a poorer prognosis. We would not have significant statistical power. We added this observation in discussion, however in prospective studies or large retrospective studies, this issue could be further studied.  

Page 5, line 210-213: we added in discussion that "Finally, no correlation between discrepancies and clinical outcome of MM patients was possible to be reported due to the limited number of patients , to the retrospective nature of the study and to the fact that the presence of focal lesion is not the only determinant of poor prognosis."

  • English should be revised in some parts

English has been largely revised.

Minor revision:

Line20 and line70: please rephrase "the accuracy in assessing accurately", this sentence is redundant

- The phrases have been corrected to avoid redundancy.

Line 49: please rephrase: " MM is a cytogenetically heterogenous disorder of clonal plasma cells known with the abbreviation "CRAB"", this sentence is not cleare, better is: " MM, known with the abbreviation "CRAB,  is a cytogenetically heterogenous disorder of clonal plasma cells".

- This phrase has been corrected, as suggested, thank you for the comment.

Reviewer 2 Report

Thank you for the opportunity to read and review this paper.

The authors present a retrospective study of 70 patients with MM who underwent CT imaging. The authors aimed to evaluate the benefit of a second reading of CT images by a specialized musculoskeletal radiologist. They conclude that a second reading by a specialized radiologist can lead to a higher detection rate of focal lesions with potential influence on patient care.

Title: “…emphasis in…” sounds wrong. Please check.

Abstract: Background – First sentence obsolete. Line 22 “solution” to what problem? Line 22 “significance”: please use a specific term, what was investigated? The increase in diagnostic accuracy? Sensitivity of lesion detection? Methods: Was the diagnosis of MM confirmed or only suspected? Were the lesions only lytic bone lesions? Or also focal non-lytic extramedullary lesions? Who did the initial reports (experience, sub-specialty training)? Results: Agreement pertaining what: per lesion agreement? Per patient? Why is osteonecrosis relevant in MM? Focal lesion: bone lesion or also extramedullary lesions? If only bone lesions were investigated, how could extramedullary lesions have influenced the results? Conclusions: line 41 “is useful to” – in what way? Does it increase diagnostic accuracy? Sensitivity? Line 41 “lesion, especially lytic”: see comment on lesions above – concise language is important.

Introduction:

a little bit excessive, especially the radiomics part, could be shortened, but all in all satisfactory. Line 72: “solution” – the authors should clearly name the problem first and introduce a possible solution afterwards. Line 75: “significance” is too vague – what was specifically investigated? Line 76f: “focus on focal lesion detection”: bone lesion? Lytic bone lesion? What does “focus on” mean exactly? e.g. what else was investigated, but not focused upon? E.g. were extramedullary lesions recorded, but not evaluated?

Materials & Methods:

Line 87 “Table 1” contains data on discrepancy scoring, not the “minimal technical standard”. What was the minimal standard? E.g. slice thickness, orientation, window settings. Oftentimes images acquired outside the hospital cannot live up to high standards used in bigger myeloma centers. Especially when looking for focal lesions between 5 – 10 mm, a slice thickness of 5 mm (which nowadays unfortunately is still commonly used) is insufficient for safe lesion detection.

Line 90ff: What did the SSO radiologists specifically interpret the scans for? Lytic bone lesions? Their size or number? Non-lytic lesions? Extramedullary manifestations? Fractures?

Line 103-108: too complicated, please rephrase.

Line 107: Agreement per patient? Per Lesion?

Line 99ff: This could be explained in a little more detail. E.g. if an initial reader recorded 15 lytic lesions, but missed the 16th, is this “an important abnormality” (“3”) or an “unimportant” one (“2”)? The authors describe “interpreting the presence of a lytic lesion > 5mm” as an example for a clinically important discrepancy, but certainly that is only relevant e.g. if the missed lytic lesion was the only one the patient has at all (turning the “B” in CRAB positive), or if the lesion was so big as to be in danger of a pathologic fracture.

In general, more could be said about the initial readers. Experience? Maybe there were some qualified musculoskeletal radiologists already involved in the initial reports, which might seriously skew the results.

Line 115ff: why include the hematologist and radiation oncologist? It looks like the authors introduce another gold standard to check the “true” gold standard (SSO – radiologists), which in turn was supposed to be compared with the initial reports? Why?

Statistics: per-patient or per-lesion agreement? Or agreement pertaining above mentioned “discrepancy scoring system”?

Results:

Table 2: So diffuse pattern was always scored as “no discrepancy”? Does that mean, that all readers (initial readers and SSO-readers) agreed on a diffuse pattern in 100% of cases? That is hard to believe, especially with the vague definition of a diffuse pattern given in line 121-122; some authors might even argue, that it is not possible to diagnose a diffuse pattern with standard CT.

Do the numbers in table 2 add up? If 46/70 reports were graded “1”, the table doesn’t reflect this. Same for “2” and “3”.

Why emphasize osteonecrosis?

Line 147: give examples or a complete list of unimportant abnormalities.

Line 150: elaborate on examples for “clinical important abnormalities”.

Line 152: Diameter of missed focal lesions? Or all detected focal lesions?

Line 153: 95%CI missing. Also give range of numbers of lesions.

In general, how many patients had focal lesions at all? How many patients had none? And in how many patients without detected lesions by the initial report did SSO-readers indeed find “new” lesions, thus potentially changing further treatment planning? I think that is the central question of this manuscript, but I do not see it addressed clearly.

Discussion:

In general, “focal lesions” detected by MRI should not be confused with “lytic lesions” detected by CT! It is critical that the authors address this fact.

Why again emphasize osteonecrosis?

Limitations: see above.

Otherwise satisfactory.

Conclusion:

Line 202: “is useful to” – in what way? Does it increase diagnostic accuracy? Sensitivity? “especially lytic”: see comment on lesions above – concise language is important, especially in the conclusion statement.

References:

Satisfactory.

Author Response

Author's Reply to the Review Report (Reviewer 2)

Comments and Suggestions for Authors

Thank you for the opportunity to read and review this paper.

The authors present a retrospective study of 70 patients with MM who underwent CT imaging. The authors aimed to evaluate the benefit of a second reading of CT images by a specialized musculoskeletal radiologist. They conclude that a second reading by a specialized radiologist can lead to a higher detection rate of focal lesions with potential influence on patient care.

Title: “…emphasis in…” sounds wrong. Please check.

- We corrected with emphasis on...it should be better.

Abstract: Background – First sentence obsolete.

- First sentence has been deleted.

Line 22 “solution” to what problem? Line 22 “significance”: please use a specific term, what was investigated? The increase in diagnostic accuracy? Sensitivity of lesion detection?

- The study is exploratory in nature, we wanted to assess if SSO is useful or not, not only assess simple diagnostic accuracy. We change the term in usefulness.

Methods: Was the diagnosis of MM confirmed or only suspected?

- Confirmed (indeed they underwent trasnplantation)

Were the lesions only lytic bone lesions? Or also focal non-lytic extramedullary lesions? Who did the initial reports - (experience, sub-specialty training)?

- Only lytic lesions. The initial report was done by general radiologists.

Results: Agreement pertaining what: per lesion agreement? Per patient? Why is osteonecrosis relevant in MM? Focal lesion: bone lesion or also extramedullary lesions? If only bone lesions were investigated, how could extramedullary lesions have influenced the results?

- Agreement was done per-lesion, we excluded osteonecrosis (not relevant) and extramedullary lesions (likely to be reported also by general radiologists).

Conclusions: line 41 “is useful to” – in what way? Does it increase diagnostic accuracy? Sensitivity? Line 41 “lesion, especially lytic”: see comment on lesions above – concise language is important.

- Formally SSO detects more focal lesions, so it could be useful. We voluntary did not assess specifically sensitivity of diagnostic accuracy because too many considerations have to be made. For example, if the study is done as "per lesion", we do not know if it is correct to say that patient care is influenced by an increased number (sensitivity increased) of focal lesion detected. We prefer to be more striking in prospective studies with patient outcome as endpoint. This study wants to demonstrate that SSO could be better than simple general radiologists assessment in MM patients, as normally happens in Radiology.

Introduction:

a little bit excessive, especially the radiomics part, could be shortened, but all in all satisfactory. Line 72: “solution” – the authors should clearly name the problem first and introduce a possible solution afterwards.

- Line 94: We softened the statement, and stated that dedicated second-opinion interpretation of medical images performed by subspecialty musculoskeletal radiologists could be useful

Line 75: “significance” is too vague – what was specifically investigated? Line 76f: “focus on focal lesion detection”: bone lesion? Lytic bone lesion? What does “focus on” mean exactly? e.g. what else was investigated, but not focused upon? E.g. were extramedullary lesions recorded, but not evaluated?

- We are now more clear and added that we wanted to investigate ... "the increased detection of focal lesions and other radiological findings of subspecialty second-opinion (SSO) consultations for CT examinations in MM patients undergoing stem cells transplantation on standard computed tomography."  

Materials & Methods:

Line 87 “Table 1” contains data on discrepancy scoring, not the “minimal technical standard”. What was the minimal standard? E.g. slice thickness, orientation, window settings. Oftentimes images acquired outside the hospital cannot live up to high standards used in bigger myeloma centers. Especially when looking for focal lesions between 5 – 10 mm, a slice thickness of 5 mm (which nowadays unfortunately is still commonly used) is insufficient for safe lesion detection.

- We apologize and added Table 1. The other Tables have been renamed.

Line 90ff: What did the SSO radiologists specifically interpret the scans for? Lytic bone lesions? Their size or number? Non-lytic lesions? Extramedullary manifestations? Fractures?

Line 144-147: we corrected and added that :" discrepancy in interpreting an important abnormality (e.g., interpreting the presence of a lytic lesion >5 mm). Lytic bone lesions, size or number, non-lytic lesions, extramedullary manifestations and osteonecrosis only if non detected by general radiologists and fractures were considered..."

Line 103-108: too complicated, please rephrase. and Line 107: Agreement per patient? Per Lesion?

Line 141-153: We corrected and simplified the whole paragraph: " Second-opinion consultation was made independently by using 3-poin scoring system. The scoring system is similar to a scoring system already published and now adapted to MM patients [12]: 1, no discrepancy; 2, discrepancy in detecting an unimportant abnormality (e.g. interpreting a bone infarct as bone island); 3, discrepancy in interpreting an important abnormality (e.g., interpreting the presence of a lytic lesion >5 mm). Lytic bone lesions, size or number, non-lytic lesions, extramedullary manifestations and osteonecrosis only if non detected by general radiologists and fractures were considered. Clinically important difference were defined as those likely to change patient care or diagnoses. For example, a lytic lesion could be used to stage the disease according to the Durie and Salmon PLUS staging system. After per lesion intra- and inter-observer agreement calculation, reports were re-evaluated together when their scorings were discordant. Discrepancies, mainly lytic lesions, that were significant enough to warrant a change in diagnosis, prognosis, invalidity (for medico-legal implications) or treatment or referral (e.g., orthopedic surgeon, radiation oncologist specialist) were recorded."

Line 99ff: This could be explained in a little more detail. E.g. if an initial reader recorded 15 lytic lesions, but missed the 16th, is this “an important abnormality” (“3”) or an “unimportant” one (“2”)? The authors describe “interpreting the presence of a lytic lesion > 5mm” as an example for a clinically important discrepancy, but certainly that is only relevant e.g. if the missed lytic lesion was the only one the patient has at all (turning the “B” in CRAB positive), or if the lesion was so big as to be in danger of a pathologic fracture.

- We agree with this comment. We tried to be clearer Line 144-157: "Second-opinion consultation was made independently by using 3-poin scoring system. The scoring system is similar to a scoring system already published and now adapted to MM patients [12]: 1, no discrepancy; 2, discrepancy in detecting an unimportant abnormality (e.g. interpreting a bone infarct as bone island); 3, discrepancy in interpreting an important abnormality (e.g., interpreting the presence of a lytic lesion >5 mm). Lytic bone lesions, size or number, non-lytic lesions, extramedullary manifestations and osteonecrosis only if non detected by general radiologists and fractures were considered. Clinically important difference were defined as those likely to change patient care or diagnoses according to suggestions given by our clinician on per patient analysis. For example, a lytic lesion could be used to stage the disease according to the Durie and Salmon PLUS staging system. After per lesion intra- and inter-observer agreement calculation, reports were re-evaluated together when their scorings were discordant. Discrepancies, mainly lytic lesions, that were significant enough to warrant a change in diagnosis, prognosis, invalidity (for medico-legal implications) or treatment or referral (e.g., orthopedic surgeon, radiation oncologist specialist) were recorded."

In general, more could be said about the initial readers. Experience? Maybe there were some qualified musculoskeletal radiologists already involved in the initial reports, which might seriously skew the results.

- We agree and we added in the Methods, line 110-115:" The initial CT reading was done by general radiologists with no known formal (ESSR Diploma, track record in MSK radiological activities) or informal (staff rounds, reports of specialized MSK exams) specialized experience in MSK radiology."

Line 115ff: why include the hematologist and radiation oncologist? It looks like the authors introduce another gold standard to check the “true” gold standard (SSO – radiologists), which in turn was supposed to be compared with the initial reports? Why?

- To be sure that the lesion is clinically important, not only a simple radiological "miss".

Statistics: per-patient or per-lesion agreement? Or agreement pertaining above mentioned “discrepancy scoring system”?

- Per lesion.  

Results:

Table 2: So diffuse pattern was always scored as “no discrepancy”? Does that mean, that all readers (initial readers and SSO-readers) agreed on a diffuse pattern in 100% of cases? That is hard to believe, especially with the vague definition of a diffuse pattern given in line 121-122; some authors might even argue, that it is not possible to diagnose a diffuse pattern with standard CT.Do the numbers in table 2 add up? If 46/70 reports were graded “1”, the table doesn’t reflect this. Same for “2” and “3”.

- Table 2 and 3, former Table 1 and 2 were completely corrected. Hope now they are more clear.

Why emphasize osteonecrosis?

- Because it requires treatment, in our centre.

Line 147: give examples or a complete list of unimportant abnormalities.

- we added that examples of unimportant abnormalities are: bone infarct as bone island, osteophytes, disc degeneration, old vertebral collapse, not neoplastic or clearly benign bone lesions

Line 150: elaborate on examples for “clinical important abnormalities”.

- an example is in Line 153: ..."for example a lytic lesion in a CT reported as negative at initial reading"...

Line 152: Diameter of missed focal lesions? Or all detected focal lesions?

- of all detected, we corrected.

Line 153: 95%CI missing. Also give range of numbers of lesions.

- we corrected in Line 216: The mean number of focal lesion per patient was 3.4 (range: 0-20; 95% C.I.:1.1-4.7).

In general, how many patients had focal lesions at all? How many patients had none? And in how many patients without detected lesions by the initial report did SSO-readers indeed find “new” lesions, thus potentially changing further treatment planning? I think that is the central question of this manuscript, but I do not see it addressed clearly.

 - We agree that this point is crucial. We clarified that in Line 216-219: "As a whole, n=60 patients had focal lesions and n=10 had none. In n=14 patients without detected lesions by the initial report SSO find “new” lesions, thus potentially changing further treatment planning."

Discussion:

In general, “focal lesions” detected by MRI should not be confused with “lytic lesions” detected by CT! It is critical that the authors address this fact.

Line 82: we added that, as suggested, "In MM, focal lesions” detected by MRI should not be confused with “lytic lesions” detected by CT."

Why again emphasize osteonecrosis?

- Because it requires treatment, in our centre.

Limitations: see above.

Otherwise satisfactory.

 Conclusion:

Line 202: “is useful to” – in what way? Does it increase diagnostic accuracy? Sensitivity? “especially lytic”: see comment on lesions above – concise language is important, especially in the conclusion statement.

 We agree: The conclusion has been modified accordingly:" In conclusion or study demonstrated that subspecialty second-opinion in multiple myeloma CT could identify lytic lesions, previously missed, amenable to influence patients care."

References:

Satisfactory.

We would like to thank the reviewer for the thorough criticism, extremely helpful to improve this manuscript. We hope to have improved the quality further.

Round 2

Reviewer 1 Report

The authors addressed all questions. Thank you.

Author Response

  • Thank you for the help in improving the manuscript further.
  • We checked again the paper with an English native speaker.

Reviewer 2 Report

I thank the authors for their corrections and additions. If not otherwise commented, changes made are sufficient.

Line 22 “solution” to what problem? Line 22 “significance”: please use a specific term, what was investigated? The increase in diagnostic accuracy? Sensitivity of lesion detection?

- The study is exploratory in nature, we wanted to assess if SSO is useful or not, not only assess simple diagnostic accuracy. We change the term in usefulness.

--> “Usefulness” still is a very vague and nondescript concept in this context: how can “usefulness” be quantified and measured? How can one prove that certain measures are “useful”? Is an increased detection rate of lytic lesions “useful”? How so? I think “usefulness” can only be proven, if specific patient-orientend/clinical endpoints (e.g. survival…) were considered. But only an increased accuracy of SSO readers is – in my opinion – in itself not a measure of “usefulness”. The authors could claim that the increased accuracy is “useful”, if they had e.g. proven that it would have resulted in a change in treatment regimen in a certain percentage of patients. But simply “detecting more lesions” is not per se “useful”.

Conclusions: line 41 “is useful to” – in what way? Does it increase diagnostic accuracy? Sensitivity? Line 41 “lesion, especially lytic”: see comment on lesions above – concise language is important.

- Formally SSO detects more focal lesions, so it could be useful.

--> I agree, but the authors can only claim that “SSO reading is more accurate”, and could have changed outcome/treatment in a certain amount of cases.

We prefer to be more striking in prospective studies with patient outcome as endpoint. This study wants to demonstrate that SSO could be better than simple general radiologists assessment in MM patients, as normally happens in Radiology.

--> That is exactly the point. “Could be better” is extremely vague. In the abstract of a scientific study, concise langue is essential. I personally like how the authors put it at the end of the introductions section: “Therefore the purpose of this study is to evaluate the increased detection of focal lesions and other radiological findings of subspecialty second-opinion (SSO) consultations for CT examinations in MM patients…” --> here they clearly state that not some vague concept of “usefulness” is evaluated, but simple metrics of diagnostic accuracy.  

Introduction:

a little bit excessive, especially the radiomics part, could be shortened, but all in all satisfactory. Line 72: “solution” – the authors should clearly name the problem first and introduce a possible solution afterwards.

- Line 94: We softened the statement, and stated that dedicated second-opinion interpretation of medical images performed by subspecialty musculoskeletal radiologists could be useful

--> see comments above about “could be useful”.

Line 75: “significance” is too vague – what was specifically investigated? Line 76f: “focus on focal lesion detection”: bone lesion? Lytic bone lesion? What does “focus on” mean exactly? e.g. what else was investigated, but not focused upon? E.g. were extramedullary lesions recorded, but not evaluated?

- We are now more clear and added that we wanted to investigate ... "the increased detection of focal lesions and other radiological findings of subspecialty second-opinion (SSO) consultations for CT examinations in MM patients undergoing stem cells transplantation on standard computed tomography."  

--> Here the authors managed to clearly phrase the purpose of this study. They should consider rephrasing other statements e.g. in the abstract along the lines of the above statement.

Author Response

Author's Reply to the Review Report (Reviewer 2)

Round 2

I thank the authors for their corrections and additions. If not otherwise commented, changes made are sufficient.

- Thank you for appreciating our efforts to improve the manusctipt.

Line 22 “solution” to what problem? Line 22 “significance”: please use a specific term, what was investigated? The increase in diagnostic accuracy? Sensitivity of lesion detection?

- The study is exploratory in nature, we wanted to assess if SSO is useful or not, not only assess simple diagnostic accuracy. We change the term in usefulness.

--> “Usefulness” still is a very vague and nondescript concept in this context: how can “usefulness” be quantified and measured? How can one prove that certain measures are “useful”? Is an increased detection rate of lytic lesions “useful”? How so? I think “usefulness” can only be proven, if specific patient-orientend/clinical endpoints (e.g. survival…) were considered. But only an increased accuracy of SSO readers is – in my opinion – in itself not a measure of “usefulness”. The authors could claim that the increased accuracy is “useful”, if they had e.g. proven that it would have resulted in a change in treatment regimen in a certain percentage of patients. But simply “detecting more lesions” is not per se “useful”.

 - We totally agree with this comment. In a prospective more complex and well-designed study we are planning to insert more robust clinical and radiological outcomes. In the present paper, we would like to highlight that SSO could be useful and worthy of further investigation. The exploratory nature of the study and the results obtained which are concordant with previous literature (on other pathological conditions) are in favour of SSO. We really agree and deeply understand your comment and we softened the statements when possible. We changed the term again in "added value (eg. increased accuracy).

Conclusions: line 41 “is useful to” – in what way? Does it increase diagnostic accuracy? Sensitivity? Line 41 “lesion, especially lytic”: see comment on lesions above – concise language is important.

- Formally SSO detects more focal lesions, so it could be useful.

--> I agree, but the authors can only claim that “SSO reading is more accurate”, and could have changed outcome/treatment in a certain amount of cases.

- We agree and il Line 41 we corrected the words as suggested. Thank you!

 We prefer to be more striking in prospective studies with patient outcome as endpoint. This study wants to demonstrate that SSO could be better than simple general radiologists assessment in MM patients, as normally happens in Radiology.

--> That is exactly the point. “Could be better” is extremely vague. In the abstract of a scientific study, concise langue is essential. I personally like how the authors put it at the end of the introductions section: “Therefore the purpose of this study is to evaluate the increased detection of focal lesions and other radiological findings of subspecialty second-opinion (SSO) consultations for CT examinations in MM patients…” --> here they clearly state that not some vague concept of “usefulness” is evaluated, but simple metrics of diagnostic accuracy.  

- Thank you for the positive comment.

Introduction:

a little bit excessive, especially the radiomics part, could be shortened, but all in all satisfactory. Line 72: “solution” – the authors should clearly name the problem first and introduce a possible solution afterwards.

- Line 94: We softened the statement, and stated that dedicated second-opinion interpretation of medical images performed by subspecialty musculoskeletal radiologists could be useful

--> see comments above about “could be useful”.

- In Line 94 we deleted the word "useful" and corrected with "accurate" as suggested

Line 75: “significance” is too vague – what was specifically investigated? Line 76f: “focus on focal lesion detection”: bone lesion? Lytic bone lesion? What does “focus on” mean exactly? e.g. what else was investigated, but not focused upon? E.g. were extramedullary lesions recorded, but not evaluated?

- We are now more clear and added that we wanted to investigate ... "the increased detection of focal lesions and other radiological findings of subspecialty second-opinion (SSO) consultations for CT examinations in MM patients undergoing stem cells transplantation on standard computed tomography."  

--> Here the authors managed to clearly phrase the purpose of this study. They should consider rephrasing other statements e.g. in the abstract along the lines of the above statement.

- We re-checked the paper and rephrased the sentences as suggested in the Abstract and in the Introduction., as suggested.

 Thank you for the positive comments.

This manuscript is a resubmission of an earlier submission. The following is a list of the peer review reports and author responses from that submission.